# Harnessing strong aromatic conjugation in low-dimensional perovskite heterojunctions for high-performance photovoltaic devices

Bo Li[1,5], Qi Liu[2,5], Jianqiu Gong[1,5], Shuai Li[1,5], Chunlei Zhang[1,5], Danpeng Gao[1], Zhongwei Chen[3], Zhen Li[1], Xin Wu[1], Dan Zhao[1], Zexin Yu[1], Xintong Li[1], Yan Wang[1], Haipeng Lu [3] ✉, Xiao Cheng Zeng [2] ✉ & Zonglong Zhu [1,4] ✉

Low-dimensional/three-dimensional perovskite heterojunctions have shown great potential for improving the performance of perovskite photovoltaics, but large organic cations in low-dimensional perovskites hinder charge transport and cause carrier mobility anisotropy at the heterojunction interface. Here, we report a low-dimensional/three-dimensional perovskite heterojunction that introduces strong aromatic conjugated low-dimensional perovskites in p-i-n devices to reduce the electron transport resistance crossing the perovskite/electron extraction interface. The strong aromatic conjugated π-conjugated network results in continuous energy orbits among $[Pb_2I_6]^{2-}$ frameworks, thereby effectively suppressing interfacial non-radiative recombination and boosting carrier extraction. Consequently, the devices achieved an improved efficiency to 25.66% (certified 25.20%), and maintained over 95% of the initial efficiency after 1200 hours and 1000 hours under ISOS-L-1I and ISOS-D-1 protocols, respectively. The chemical design of strong aromatic conjugated molecules in perovskite heterojunctions provides a promising avenue for developing efficient and stable perovskite photovoltaics.

Organic–inorganic metal halide perovskite solar cells (PSCs) are a promising photovoltaic technology, with certified power conversion efficiency (PCE) exceeding 26%[1–4]. However, the commercialization of PSCs faces a significant challenge in terms of their stability[5]. The degradation of three-dimensional (3D) perovskites is primarily attributed to the high density of defects and ion migration at grain boundaries and interfaces, particularly under high operating temperatures[6–9]. The growth of low-dimensional (LD) perovskite layers on the surface of 3D perovskites, forming a perovskite heterojunction, offers an effective approach to passivate surface defects and suppress-ion migration, thus greatly enhancing the stability and PCE of PSCs[10–15] However, LD structures based on large cations often exhibit poor charge transport across the organic layer due to two main factors: (i) low carrier mobility in organic layers and (ii) energy barriers between inorganic frameworks and organic cations[16]. Therefore, achieving a balance between stability and charge transport efficiency is crucial in the design of LD/3D perovskite heterojunction.

Here, we show a rational design to optimize the electron transport of LD perovskites in heterojunction by introducing strong aromatic conjunction (SAC) molecules as shown in Fig. 1a. In conventional LD/3D perovskite heterojunction, free carriers are located at the conduction band minimum (CBM) and valence band maximum (VBM) of the inorganic Pb–I framework, respectively, and the out-of-plane charge transport has to go through a large cationic organic layer due to the long distance between two adjacent inorganic planes[17–19]. To overcome this limitation, conjugated LD perovskites are introduced, as

[1]Department of Chemistry, City University of Hong Kong, Kowloon, Hong Kong, China. [2]Department of Materials Science & Engineering, City University of Hong Kong, Kowloon, Hong Kong, China. [3]Department of Chemistry, The Hong Kong University of Science and Technology, Clear Water Bay, Kowloon, Hong Kong, China. [4]Shenzhen Research Institute, City University of Hong Kong, Shenzhen 518057, China. [5]These authors contributed equally: Bo Li, Qi Liu, Jianqiu Gong, Shuai Li, Chunlei Zhang. ✉e-mail: haipenglu@ust.hk; xzeng26@cityu.edu.hk; zonglzhu@cityu.edu.hk

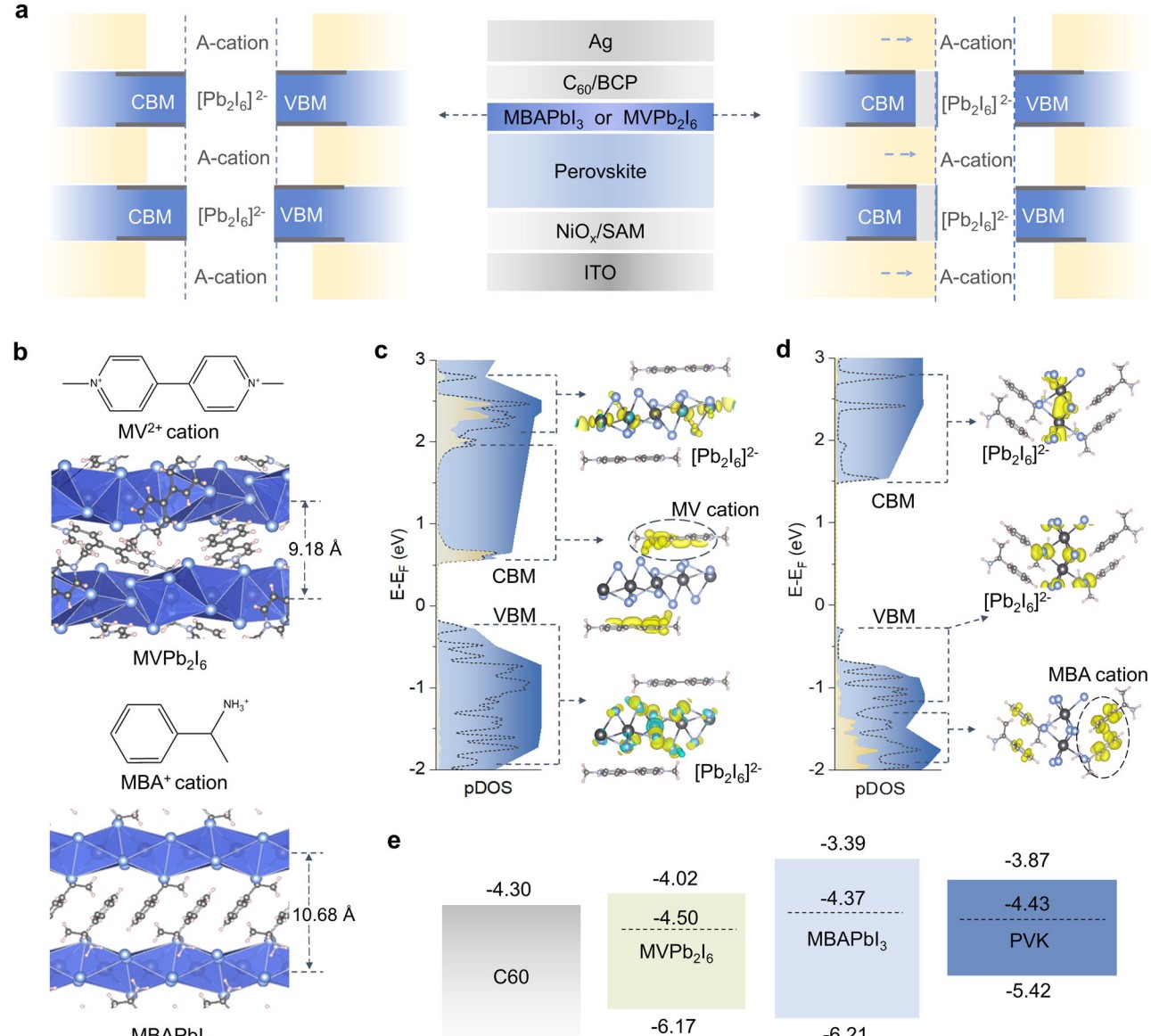

**Fig. 1 | Design concept of SAC-based heterojunction. a** Schematic structural diagram of PSCs with LD perovskite layer and the band offsets between Pb–I frameworks and A-site organic cations in LD perovskites including MVPb₂I₆ (right) and MBAPbI₃ (left). The blue dashed lines are the relocated conduction band and valance band edge. The shaded sections in right panel denote the band offset due to the replacement of MBA by MV cations. CBM and VBM represent conduction band minimum and valence band maximum respectively. **b** Optimized schematic crystal structures of 1D MVPb₂I₆ and MBAPbI₃ perovskites. **c** pDOS diagram of the MVPb₂I₆ perovskite with emphasizing the CBM, VBM and MV²⁺ electronic contributions, **d** pDOS diagram of the MBAPbI₃ perovskite with emphasizing the CBM, VBM and MBA⁺ electronic contributions. The charge density (yellow and cyan) highlights the contributions of the pDOSs in (**c**, **d**) chart. **e** Energy level diagram of the ETL/perovskite interface, including 3D perovskite, 1D MVPb₂I₆ and MBAPbI₃, as well as C₆₀. Source data are provided as a Source Data file.

they consist of aromatic rings to form π-bond networks, where π orbitals can form continuous energy bands, enabling carriers to shuttle more efficiently between the inorganic Pb–I framework and the cationic organic layer[20–24]. In addition, the strong π-π stacking interaction between the aromatic rings of SAC molecules endows the perovskites with superior thermal and chemical stability[25].

Based on this design, we produce a SAC-based LD/3D perovskite heterojunction by growing MVPb₂I₆ on the surface of 3D perovskite using N, N′-dimethyl-4,4-bipyridinium iodide (MVI₂) solution. The LD/3D perovskite heterojunction based on SAC molecule achieve a suppressed non-radiative recombination and boosted electron extraction due to the reduction of perovskite surface defects and organic cation barrier. The resulting PSCs exhibit an enhanced PCE up to 25.66%, and retain over 95% of the initial PCE under ISOS-L-1I and ISOS-D-1 protocols for 1200 h and 1000 h, respectively. Additionally, the energy loss

analysis reveals that the SAC-based LD layer results in a decrease of around 34% and 14% for non-radiative recombination and transport loss, respectively, proving the great potential of SAC molecules in LD/3D heterojunction for efficient PSCs.

## Results

### Structure and electronics of SAC-based LD/3D heterojunction

We investigated the perovskite-C₆₀ interface in p–i–n based PSCs as it is the most severe culprit for inducing trap states and causing device instability[26]. The device structure is shown in Fig. 1a, where nickel oxide and self-assembled monolayer (NiOₓ-SAM) and C₆₀ are used as p-type and n-type charge extraction layers, respectively. The MVPb₂I₆ perovskite layer is formed by depositing MVI₂ solution on the 3D perovskite surface through in-situ chemical reaction. The crystal structure of MVPb₂I₆ perovskite is shown in Fig. 1b and

Supplementary Fig. 1, where cation-π conjugated $MV^{2+}$ cations are electrostatically combined with $[Pb_2I_6]^{2-}$ to form a hybrid one-dimensional (1D) structure[27–30]. These 1D $[Pb_2I_6]^{2-}$ chains form a coplanar octahedral inorganic framework, which is separated by a uniform distance of 9.18 Å and surrounded by cross-aligned MV cations. As a comparison, we chose methylbenzylammonium ($MBA^+$) to prepare $MBAPbI_3$[31], which has similar 1D $[Pb_2I_6]^{2-}$ chains as $MVPb_2I_6$ but varied chain spacing (in Fig. 1b and Supplementary Fig. 1), in order to compare the effects of different heterojunctions on the performance of PSCs.

The $MVPb_2I_6$ and $MBAPbI_3$ single crystals were synthesized by the antisolvent vapor-assisted crystallization method and their structures and the variations of organic cations are illustrated in Supplementary Figs. 2–5. To construct the LD/3D heterojunction, we deposited the SAC-containing organic cation solution on the 3D perovskite films and annealed them to grow the 1D structure in-situ on their surface. For the X-ray diffraction (XRD) patterns in Supplementary Figs. 6 and 7, as the concentrations of MV and MBA containing cation solution increase, the characteristic peaks of 11.1° for 1D $MVPb_2I_6$ and 7.1° (8.3°) for 1D $MBAPbI_3$ in LD/3D heterojunction films become more pronounced. Furthermore, for the film morphologies determined by scanning electron microscopy (SEM) and atomic force microscopy (AFM) in Supplementary Figs. 8 and 9, the surface roughness of the films is reduced, accompanied by the blurring of the crystal boundaries after the formation of LD/3D heterojunction. The changes of the crystal structures and morphologies suggest the formation of LD $MVPb_2I_6$ and $MBAPbI_3$ on the 3D perovskite. X-ray photoelectron spectroscopy (XPS) was used to further characterize the chemical valence state changes on the film surface. The introduction of $MV^{2+}$ cation lowers the binding energy of Pb $4f$ and I $3d$, suggesting that the electron density around Pb and I increases due to the strong conjugation of MV (Supplementary Figs. 10 and 11). Additionally, the N $1s$ spectrum of $MVPb_2I_6$/PVK films shows a new C=N–C peak from pyridinium, further confirming the formation of LD/3D heterojunction (Supplementary Fig. 12).

The $MVPb_2I_6$ shows a broader absorption spectrum from ultraviolet to nearly 680 nm compared to that of $MBAPbI_3$ in Supplementary Fig. 13. The broadening of the absorption range can be attributed to interfacial charge transfer absorption due to the strong aromatic interaction of $MV^{2+}$ cations and the unique $MVPb_2I_6$ structure[32–35]. Moreover, the mobility of $MVPb_2I_6$ is nearly an order of magnitude faster than that of $MBAPbI_3$ in Supplementary Note 1 and Supplementary Fig. 14. We performed Density Functional Theory (DFT) calculations to analyze their electronic structures in Supplementary Figs. 15 and 16, and the corresponding projected density of states (pDOS) are shown in Fig. 1c, d. The VBM and CBM of $MVPb_2I_6$ perovskite are located at −0.1 and 0.5 eV, respectively, where the $[Pb_2I_6]^{2-}$ chains mainly contribute to the VBM and yield localized states in the deep conduction band, while the MV cation contributes to the CBM of $MVPb_2I_6$ perovskite. This band structure indicates that the MV cation forms large π-bond networks that can contribute continuous states at the CBM, greatly lowering the CBM level contributed by isolated 1D $[Pb_2I_6]^{2-}$ chains. Moreover, the formation of large π-bond networks at the CBM endows electrons with higher mobility. In contrast, the CBM and VBM of the $MBAPbI_3$ are located at 1.5 eV and −0.2 eV, respectively, both contributed by $[Pb_2I_6]^{2-}$ chains, while the $MBA^+$ cation only contributes some states in the deep valence band. This gives rise to a wide bandgap and an energy barrier between the $[Pb_2I_6]^{2-}$ and the organic cation, limiting electron transport in LD perovskite structures. These changes in electronic structure are also confirmed by ultraviolet photoelectron spectroscopy (UPS) in Supplementary Fig. 17, where the $MVPb_2I_6$ has a lower CBM than $MBAPbI_3$, resulting in a type-II alignment at the interface with 3D perovskite that facilitates electron transfer from the perovskite layer to the electron transport layer (Fig. 1e).

## Carrier and defects in SAC-based LD/3D heterojunction

To study the charge carrier dynamics in LD/3D heterojunction, we performed conductive atomic force microscopy (C-AFM) and photo-luminescence (PL) mapping analysis. In Fig. 2a, the control perovskite (PVK) without heterojunction shows a current of 140 pA, which is similar to that of the $PVK/MVPb_2I_6$ as shown in Fig. 2c, indicating that the $MVPb_2I_6$ layer does not affect out-of-plane carrier transports. In contrast, the $PVK/MBAPbI_3$ exhibits a reduced current of about 70 pA, suggesting a high electron resistance of $MBAPbI_3$ layer[36]. Fig. 2b shows the PL maps of perovskite films with $C_{60}$ to estimate the electron extraction efficiency. Compared with the control PVK, $PVK/MVPb_2I_6$ exhibits reduced and homogenized PL map intensity in Fig. 2d, indicating accelerated and uniform electron extraction at the interface, while $PVK/MBAPbI_3$ exhibits the opposite trend. The time-resolved PL spectra in Fig. 2e also confirm this trend, where the $PVK/MVPb_2I_6$ film shows a reduction of carrier lifetime from 22.04 ns to 12.09 ns (Supplementary Table 1). These results indicate that the modification of $MVPb_2I_6$ can overcome the out-of-plane charge transport barrier caused by organic cations to enhance the interface electron extraction, by forming strong aromatic conjunction electron transport channels.

Kelvin probe force microscopy (KPFM) was used to study the surface potential changes of the LD/3D heterojunction. In Fig. 2f, the control film exhibits an inhomogeneous surface potential, with potential ranging from 0.5 to 1.3 V, which is caused by the surface and grain boundary defects of the perovskite film[37]. After the $MBA^+$ and $MV^{2+}$ cation modification, the film surface potentials tend to be homogenized. Moreover, in Fig. 2g, the $PVK/MVPb_2I_6$ shows a narrower potential range (from 0.7 to 1.0 eV) and smaller potential offsets between the grain bulk and boundary than $PVK/MBAPbI_3$, indicating that the defects on the surface and at the grain boundaries of the perovskite film are suppressed. Steady-state PL and time-resolved PL spectra are applied to further confirm the defect passivation effects in Supplementary Figs. 18 and 19. The $PVK/MVPb_2I_6$ film displays a significant increase in PL intensity compared to the control and $PVK/MBAPbI_3$ films. Moreover, the carrier lifetime enhances from 341.17 ns to 1273.22 ns as the $PVK/MVPb_2I_6$ heterojunction formation (Supplementary Table 2). The increasements of PL intensity and lifetime imply improved reduced defect density and non-radiative recombination[38].

## Photovoltaic performance and stability

We studied the impact of LD/3D heterojunction on the photovoltaic performance of PSCs. The cross-sectional scanning electron microscopy (SEM) image of the typical device structure is shown in Supplementary Fig. 20. The performance of the PSCs based on the control PVK, $PVK/MBAPbI_3$ and $PVK/MVPb_2I_6$ with different concentrations was optimized and shown in Supplementary Figs. 21–24, and Supplementary Tables 3–5. Figure 3a compares the current–voltage ($J$–$V$) characteristics of the best-performing PSCs of these devices under simulated AM 1.5G illumination. The control device shows a maximum PCE of 23.32%, with an open-circuit voltage ($V_{OC}$) of 1.112 V, a short-circuit current density ($J_{SC}$) of 25.41 mA cm$^{-2}$ and a fill factor (FF) of 82.54% (Supplementary Table 3). With the surface treatment, the $PVK/MBAPbI_3$ and $PVK/MVPb_2I_6$ devices exhibit enhanced $V_{OC}$ and FF values. Especially, the $PVK/MVPb_2I_6$ device achieves a remarkable PCE of 25.66% from the forward scan and 25.57% from the reverse scan as shown in Fig. 3b, which ranks among the best-performing LD/3D based PSCs (Supplementary Table 6). One of the best-performance PSCs was sent to an independent certification laboratory (SIMIT, China), where a PCE of 25.20% was confirmed (Supplementary Fig. 25).

The external quantum efficiency (EQE) spectrum of the champion device is shown in Fig. 3c, which agrees well with the J-V measurements, with a minor difference in the integrated current values. Figure 3d and Supplementary Fig. 26 display the photovoltaic parameter statistics of the control PVK, $PVK/MBAPbI_3$ and $PVK/MVPb_2I_6$ devices. It is noteworthy that the devices modified with MBAI only show a

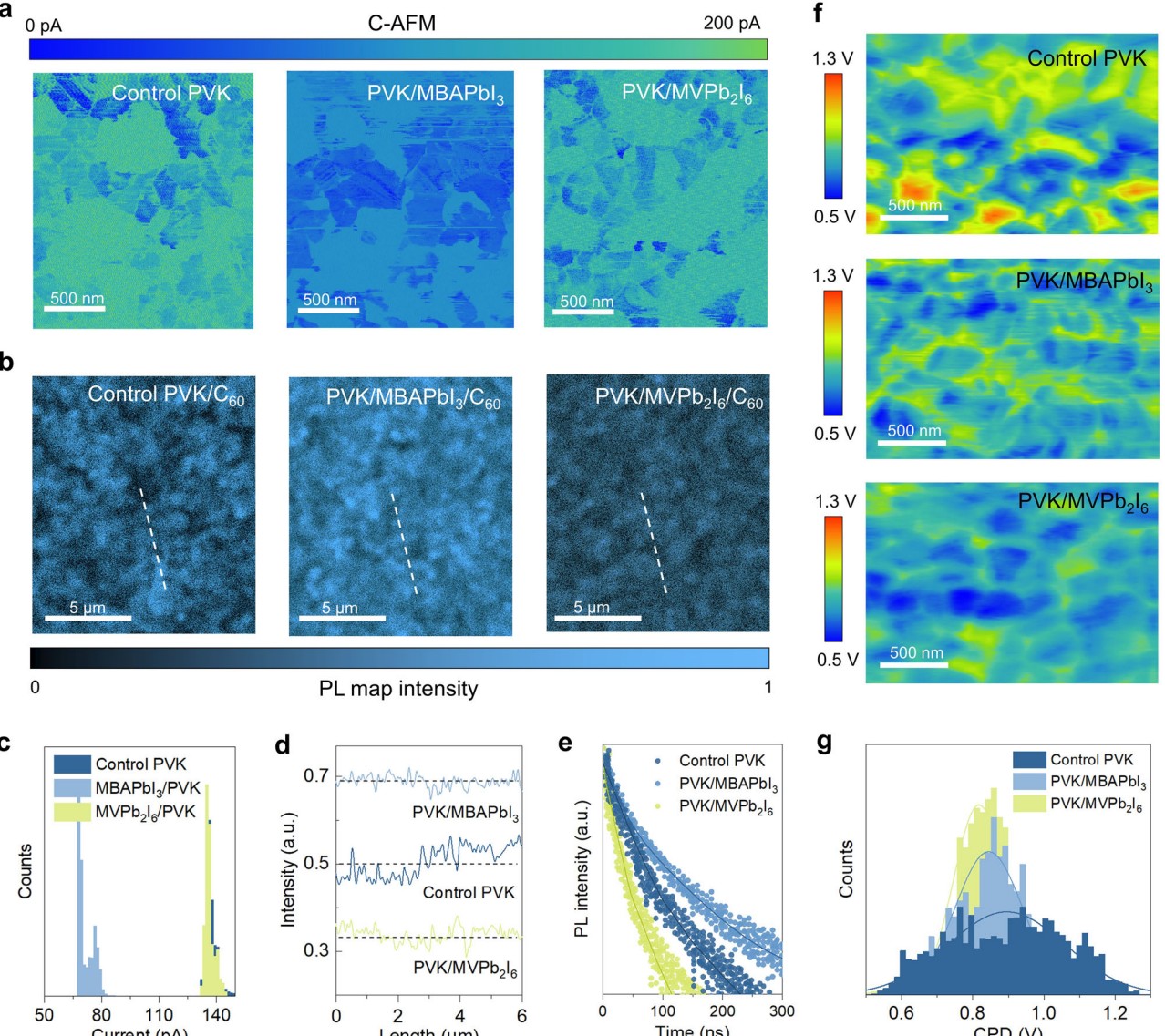

**Fig. 2 | Carrier dynamics and defects characterization. a** C-AFM images of the control PVK, PVK/MBAPbI$_3$ and PVK/MVPb$_2$I$_6$ films. **b** PL maps of the control PVK/ C$_{60}$, PVK/MBAPbI$_3$/C$_{60}$ and PVK/MVPb$_2$I$_6$/C$_{60}$ films. **c** Statistics of current intensities extracted from the C-AFM images. **d** PL intensities extracted from the dashed lines in PL maps. **e** TRPL spectra of the control PVK/C$_{60}$, PVK/MBAPbI$_3$/C$_{60}$ and PVK/ MVPb$_2$I$_6$/C$_{60}$ films. **f** KPFM images of the control PVK, PVK/MBAPbI$_3$ and PVK/ MVPb$_2$I$_6$ films. **g** Statistics of surface potential difference (CPD) extracted from KPFM images. The incident excitation light enters from the C$_{60}$ side for PL and TRPL measurements. Source data are provided as a Source Data file.

noticeable improvement in $V_{OC}$, while the devices modified with MVI$_2$ exhibit a remarkable increase in $V_{OC}$ compared to other devices, as well as slight enhancements in $J_{SC}$ and FF. In addition, the stabilized output of the PVK/MVPb$_2$I$_6$ device was tested in Fig. 3e and the stable current density and power output were obtained as 24.21 mA cm$^{-2}$ and 25.18%, respectively.

We evaluated the stability of the PSCs based on the protocols described in the consensus report[39]. Fig. 3f shows the light-soaking stability (ISOS-L-1I) of the encapsulated devices under normal working conditions with maximum power-point (MPP) tracking in N$_2$. The devices based on PVK/MVPb$_2$I$_6$ retain over 95% of initial PCE after operating continuously for 1200 h. In contrast, the control and the PVK/MBAPbI$_3$ devices maintain only 85.4% and 83% of their initial PCE after 1200 h and 800 h, respectively. Under the ISOS-D-1 protocol in Fig. 3g, the PCE of the PVK/MVPb$_2$I$_6$ devices drops to 95.3% of the initial value after 1000 h, whereas the control and the PVK/MBAPbI$_3$ devices show over 20% degradation. The enhanced stability after introducing

MVPb$_2$I$_6$ LD layer can be attributed to their excellent stability against polar solvent (Supplementary Fig. 27) as well as their improved interface charge extraction and reduced surface defect density.

### Energy loss analysis
We performed the energy loss analysis to study how the inclusion of SAC heterojunction affects the photovoltaic performance of PSCs. Figures 4a to 4c show electroluminescence (EL) spectra of the devices based on the control, PVK/MBAPbI$_3$ and PVK/MVPb$_2$I$_6$. The EL intensities are normalized to the highest value of all spectra. The devices with MVPb$_2$I$_6$ layer exhibit higher emission intensity than those of the control and the PVK/MBAPbI$_3$ devices. Furthermore, the PVK/MVPb$_2$I$_6$ device displays a narrower emission range compared to that of the control one, which suggests the suppression of non-radiative recombination[40]. As a result, in Fig. 4d, the PVK/MVPb$_2$I$_6$ device demonstrates an 8.2% EQE-EL efficiency. In contrast, the control and PVK/MBAPbI$_3$ devices only show 3.2% and 5.4% EQE-EL efficiency under the same conditions.

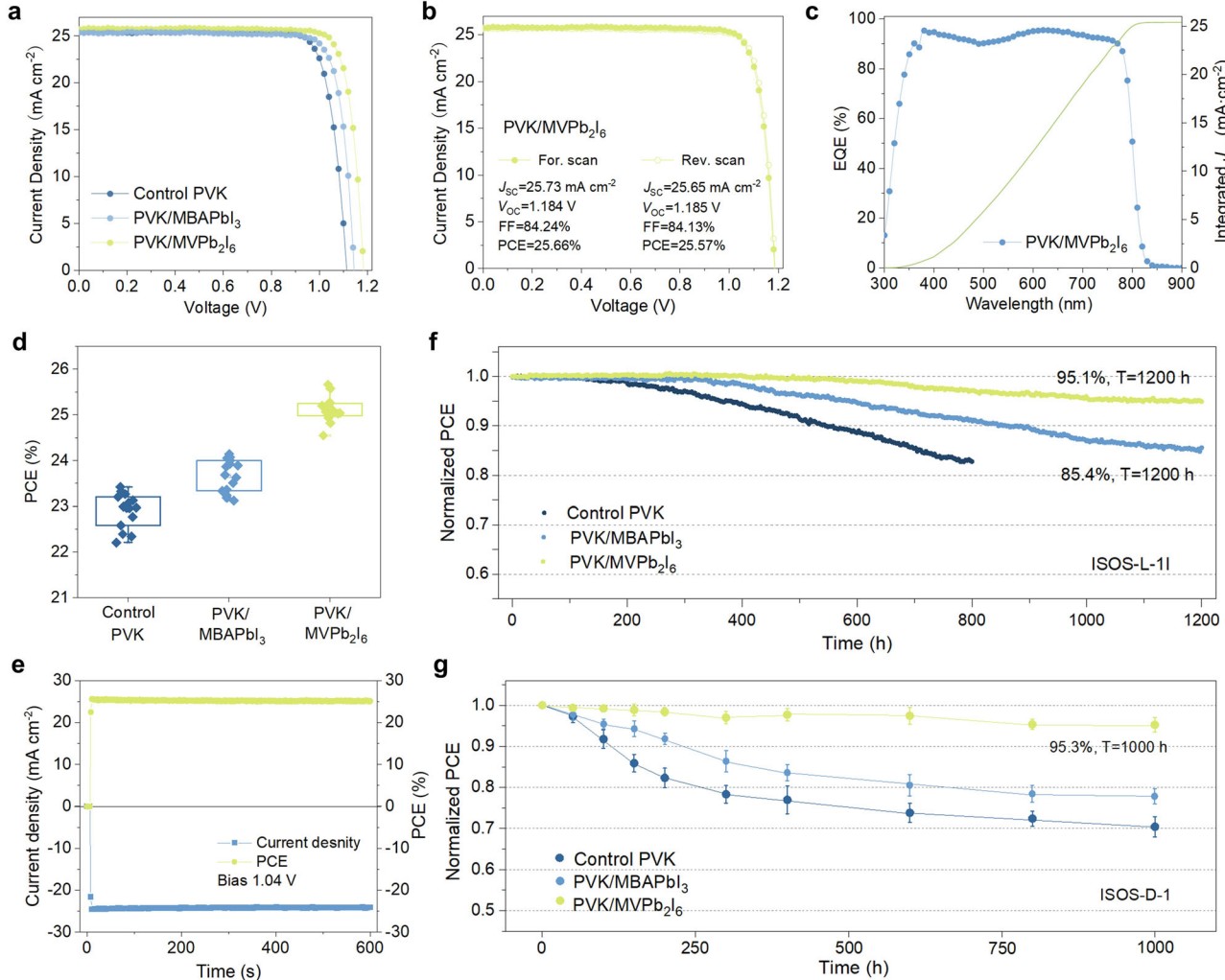

**Fig. 3 | Photovoltaic performance and stability. a** $J$–$V$ curves of the best-performing PSCs with the control PVK, PVK/MBAPbI$_3$ and PVK/MVPb$_2$I$_6$. **b** $J$–$V$ curves of the best-performing PVK/MVPb$_2$I$_6$ PSCs with forward and reverse scans. **c** EQE curves and integrated $J_{SC}$ of the PVK/MVPb$_2$I$_6$ PSC. **d** Statistics of the PCEs among 15 individual devices with different heterostructures. **e** Stabilized current and efficiency output under continuous illumination. **f** ISOS-L-1I stability of the encapsulated devices measured at MPP under continuous AM 1.5 G simulated solar illumination at room temperature in N$_2$. **g** ISOS-D-1 stability of the unencapsulated device measured at room temperature and 20–40% relative humidity. Error bar represents the confidence interval of each stability. Source data are provided as a Source Data file.

The $V_{OC}$ loss was quantitatively analyzed based on the detailed balance theory[41], in which the loss can be divided into three parts, namely, radiative recombination above E$_g$ ($\Delta V_1$), black-body radiation loss ($\Delta V_2$) and non-radiative recombination loss ($\Delta V_3$) in Fig. 4e. The detailed analysis and loss results are shown in Supplementary Note 2 and Supplementary Table 7 respectively. The $\Delta V_2$ based on highly sensitive EQE spectra (Supplementary Fig. 28) decreases from 84.41 mV for the control to 29.86 mV for the PVK/MVPb$_2$I$_6$. The reduced $\Delta V_2$ can be attributed to reduced sub-bandgap EQE caused by the suppression of energy disorder[42,43]. The $\Delta V_3$ value declines from 88.98 mV to 64.65 mV after the introduction of MVPb$_2$I$_6$, implying effective alleviation of non-radiative recombination. Trap density of states (tDOS) of the devices was also measured with thermal admittance spectroscopy. In Fig. 4f, the device based on PVK/MVPb$_2$I$_6$ shows a lower tDOS compared to other devices, especially in the deep-trap state region, which indicates that the reduced photovoltage loss mainly originates from the suppression of deep-trap states[44].

We further analyzed the FF loss of PSCs. Figure 4g shows the voltages of PSCs measured at different light intensities (described in Supplementary Note 3), from which the ideality factors can be extracted as 1.45, 1.34 and 1.26 for the control, PVK/MBAPbI$_3$ and PVK/MVPb$_2$I$_6$, respectively. We obtained the statistics of FF loss as shown in Fig. 4h, including the contributions of transport loss and non-radiative recombination loss, based on the Shockley-Queisser limit of FF and ideality factor (Supplementary Note 4). The devices with MBAPbI$_3$ and MVPb$_2$I$_6$ exhibit gradually reduced non-radiative loss, with decrease rates of 25% and 34%, respectively, which can be attributed to the suppression of deep level trap states in Fig. 4i. Moreover, this reduction is consistent with the $V_{OC}$ loss calculation in Fig. 4e. In addition, compared to the charge transport loss of the control device, the device based on PVK/MVPb$_2$I$_6$ displays smaller value, while the opposite trend is observed for PVK/MBAPbI$_3$. This change indicates that MVPb$_2$I$_6$ LD layer has faster interfacial charge transfer efficiency than MBAPbI$_3$, which is due to the strong aromatic conjugation of MV cation in Fig. 4i. This result is also in agreement with the electrochemical impedance spectra (EIS) test in Supplementary Fig. 29.

## Discussion

In summary, we develop a LD/3D heterojunction interface by introducing strong aromatic conjugated LD perovskite materials, which can establish an electron transport channel between the [Pb$_2$I$_6$]$^{2-}$

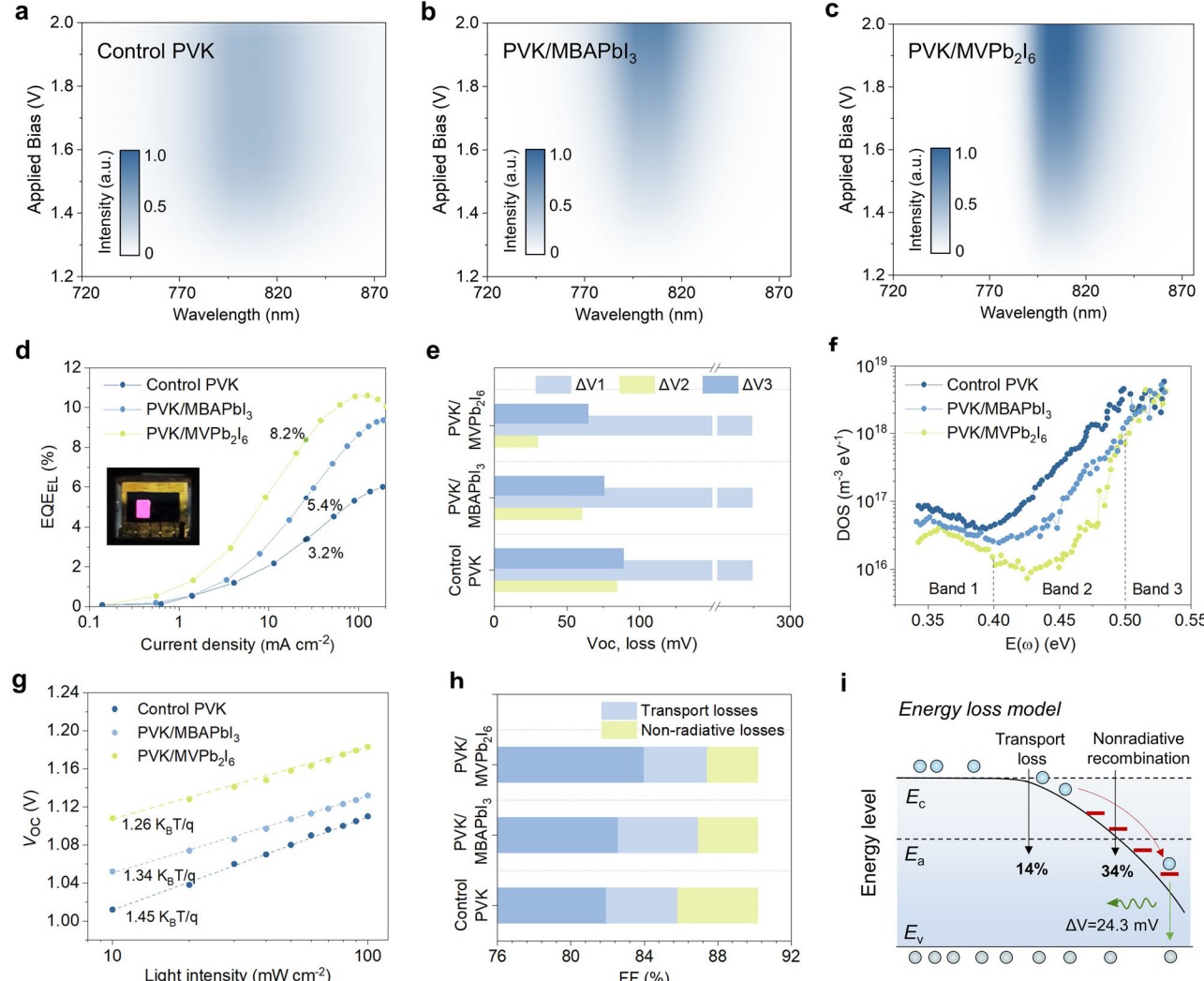

**Fig. 4 | Energy loss analysis. a–c** EL spectra of the PSCs based on the control PVK, PVK/MBAPbI₃ and PVK/MVPb₂I₆ under different applied voltage bias. **d** EQE-EL spectra of the PSCs operating in light-emitting diodes. **e** $V_{OC}$ loss calculation in the devices based on the control PVK, PVK/MBAPbI₃ and PVK/MVPb₂I₆, including radiative recombination above $E_g$ ($\Delta V_1$), black-body radiation loss ($\Delta V_2$) and non-radiative recombination loss ($\Delta V_3$). **f** tDOS curves of the three devices, in which 0.35 to 0.40 eV is a shallow trap state (band 1), and >0.4 eV is a deep-trap state (band 2 and band 3). **g** Light intensity dependent $V_{oc}$ values. **h** FF loss calculation in the devices based on the control PVK, PVK/MBAPbI₃ and PVK/MVPb₂I₆. **i** Schematic illustration of the interface energy loss in the PVK/MVPb₂I₆ based PSCs. Source data are provided as a Source Data file.

frameworks to break the transport barrier of the organic layer. This surface engineering strategy results in an efficiency of 25.66% and over 95% efficiency retention after 1200 and 1000 h of stability tests based on ISOS-L-1I and ISOS-D-1 protocols. Carrier dynamics and defect state analysis confirm the marked contribution of the SAC-based LD perovskite layer in facilitating charge transfer and suppressing defect states. Energy loss analysis quantifies the device losses in transport and non-radiative recombination. The chemistry design of the strong aromatic conjugated LD perovskite materials opens a new avenue for developing LD/3D heterojunction to achieve highly efficient and stable PSCs.

## Methods
### Materials
Formamidinium iodide (FAI), methylammonium bromide (MABr), methylammonium chloride (MACl) and cesium iodide (CsI) were purchased from Dysol (Australia). Lead iodide (PbI₂), lead bromide (PbBr₂) and [2-(3,6-Dimethoxy-9H-carbazol-9-yl)ethyl]phosphonic Acid (MeO–2PACz) were purchased from TCI (Japan). C₆₀, bathocuproine (BCP, purity of 99.9%) and (R)-(+)-α-methylbenzylamine

iodide (R-MBAI) were purchased from Xi'an Polymer Light Technology Corporation (China). Nickel oxide particle (NiOₓ) was purchased from Advanced Election Technology Co., Ltd (China), the average nanoparticle size is 7 nm. Solvents, including dimethylformamide (DMF), dimethyl sulfoxide (DMSO), isopropanol (IPA) and chlorobenzene (CB) were purchased from J&K (China) and used as received. 4,4-bipyridine (C₁₀H₈N₂, 98%) was purchased from Aladdin. High purity silver was purchased from commercial sources. was purchased from TCI (Japan). Glass substrates patterned with indium tin oxide (ITO) (15 Ω sq⁻¹) were received from Mishi Tech. Co., Ltd (China).

### Single crystal growth
For synthesis of MVPb₂I₆ and MBAPbI₃: PbI₂ (272 mg) and MVI₂ (128 mg) or MBAI (71.5 mg) were dissolved in DMSO (3 mL) and stirred overnight at 50 °C. The turbid crimson liquid was filtered and transferred into a 4 mL vial. The vial was then placed into a sealed bottle filled with 2 mL CB. The single crystals were grown along with the slow diffusion of the vapor of the antisolvent CB into the solution. The MVI₂ was prepared according to the reported method[45].

## LD perovskite film fabrication

For $MVPb_2I_6$: 38 mg $MVI_2$ and 81 mg $PbI_2$ were dissolved in 1 ml DMSO and filtered to obtain a clear solution. Before spin-coating, the substrate and solution were preheated on a hotplate at 70 °C for 5 min. The solution was then immediately spin-coated onto the substrate at 3000 rpm for 30 s, and the resulting film was annealed at 150 °C for 10 min. For $MBAPbI_3$: 21.3 mg MBAI and 81 mg $PbI_2$ were dissolved in 1 ml DMSO. Before spin-coating, the substrate and solution were preheated on a hotplate at 70 °C for 5 min. The solution was then immediately spin-coated onto the substrate at 3000 rpm for 30 s, and the resulting film was annealed at 100 °C for 10 min.

## Device fabrication

ITO substrates were sequentially cleaned with detergent, deionized water, acetone and isopropyl alcohol under ultrasonication for 20 min, respectively. Then, the ITO substrates were dried and treated with oxygen plasma for 20 min. $NiO_x$ was dispersed in DI water with a concentration of 10 mg/mL by ultrasonication for 10 min and filtered before using. 50 μL $NiO_x$ solution was spin-coated onto ITO substrates at 4000 rpm for 30 s, annealed at 150 °C for 20 min in ambient atmosphere, and then transferred into a $N_2$-filled glovebox. 0.3 mg/mL Meo-2PACz was dissolved in EtOH. 80 μL as-prepared Meo-2PACz solution was spin-coated onto the $NiO_x$ layer and annealed on a hotplate at 100 °C for 10 min.

For the control perovskite. The perovskite solution (1.73 M) was prepared by mixing CsI, FAI, MABr, $PbI_2$ and $PbBr_2$ in 1 ml mixed DMF:DMSO (5:1/v:v) solvent for a chemical formula $(FA_{0.98}MA_{0.02})_{0.95}Cs_{0.05}Pb(I_{0.98}Br_{0.02})_3$, 5 mol% of excess $PbI_2$ was needed to improve the device performance. Then 15.5 mol% MACl was added to the perovskite precursor solution and stirred for 2 h. 80 μL perovskite solutions were spin-coated onto ITO/MeO-2PACz at 1000 rpm for 10 s, subsequently at 5000 rpm for 40 s. 350 μL CB was dripped onto the center of film at 12 s before the end of spin-coating. The deposited perovskite films were subsequently annealed on a hotplate at 100 °C for 20 min.

For the MV and MBA based LD/3D perovskite. The spin-coating processes were all conducted at room temperature in a $N_2$-filled glovebox with the contents of $O_2$ and $H_2O < 10$ ppm. $MVI_2$ and MBAI are spin-coated onto perovskite in the same way. They are dissolved in a mixed solution of Methanol and DMF (150:1) at a certain concentration, and heated to 60 °C for 20 min until the solution became clear. 80 μL as-prepared $MVI_2$ or MBAI solution was spin-coated onto the perovskite at 5000 rpm for 30 s, and annealed on a hotplate at 100 °C for 10 min.

Finally, 20 nm C60 at a rate of 0.5 Å s$^{-1}$, 6 nm BCP a rate of 0.5 Å s$^{-1}$ and 100 nm silver electrode a rate of 1.0 Å s$^{-1}$ were thermally evaporated, respectively, under high vacuum (<4 × 10$^{-6}$ Torr).

## Characterization

Surface and cross-section morphologies of the perovskite films were examined using scanning electron microscopy (SEM) on a QUATTROS instrument from Thermal Fisher Scientific. X-ray diffraction (XRD) measurements were performed using X-ray diffractometer (Rigaku SmartLab system) with Cu Kα radiation. X-ray photoelectron spectroscopy (XPS) measurements were performed using an AXIS Supra XPS system. Steady-state and time-resolved photoluminescence (PL) spectra were obtained using an Edinburgh FLS980 instrument with an excitation wavelength of 485 nm. Ultraviolet–visible (UV–vis) absorption spectra were recorded using a Perkin Elmer model Lambda 2S spectrometer. The film thickness was measured using a DektakXT stylus profiler. All atomic force microscopy (AFM)-based characterizations, including conductive AFM and Kelvin probe force microscopy, were conducted using a SHIMADZU SPM-9700HT system. The tDOS was obtained from the thermal admittance spectroscopy (TAS) on an LCR meter (Agilent E4980A).The photovoltaic performance of the perovskite solar cells was evaluated using a Xenon lamp solar simulator (Enlitech, SS-F5, Taiwan) in a $N_2$-filled glovebox at room temperature. The light intensity was calibrated to 100 mW cm$^{-2}$ using a silicon reference cell with a KG2 filter. Prior to the $J$–$V$ measurements, a 120-nm thick magnesium fluoride layer was deposited on the back of the ITO substrate to enhance transmittance. $J$–$V$ curves were acquired using a Keithley 2400 source meter under a reverse scan sweep mode (from 1.20 V to −0.01 V) and a forward scan sweep mode (from −0.01 V to 1.20 V), with a scan rate of 0.01 V s$^{-1}$ and a delay time of 10 ms. The active area of the devices was defined as 0.0414 cm$^2$ for small-area devices using a metal shadow mask. The stabilized power output (SPO) was determined by monitoring the stabilized current density output at the maximum power point (MPP) bias, which was extracted from the reverse scan $J$–$V$ curves. External quantum efficiency (EQE) measurements were performed using a QE-R EQE system (Enlitech, Taiwan). Electrochemical impedance spectroscopy (EIS) was measured with a ZAHNER Zennium Electrochemical Workstation in the frequency range of 100 mHz to 1 MHz under open-circuit and dark condition.

## Stability tests

The ISOS-L-1I stability was conducted by applying the perovskite solar cells under a 1 sun equivalent LED lamp in a $N_2$-filled glovebox (with the contents of $O_2$ and $H_2O < 10$ ppm) at room temperature. The PSCs were biased at maximum-power-point (MPP) voltage and the power output was tracked by using a multi-potentiostat (CHI1040C, CH Instruments, Inc.). During the MPP test, the current density-voltage ($J$-$V$) curves of the devices were obtained every 12 h to get the proper loads for the MPP. The ISOS-D-1 stability was conducted to study the evolution of normalized PCE for non-encapsulated solar cells aged at room temperature of RH of 20-40% in the dark.

## Density functional theory (DFT) calculations

The first-principles DFT simulations were performed with the Vienna Ab Initio Simulation Package (VASP 6.4)[46–48] to study the geometric and electronic structures of all the perovskite series. Unless otherwise specified, the projector augmented wave (PAW) pseudopotentials with the cutoff energy of 600 eV were employed[47]. The generalized gradient approximation (GGA) exchange-correlation functional of Perdew-Burke-Ernzerhof (PBE) was adopted in the DFT calculations[48]. The spin-orbit coupling (SOC) effect was adopted in the electronic structure calculations of the perovskite frameworks. The electronic constituents are $5d$ $6s$ $6p$ for P$b$, 5p $6s$ for I, $2s$ 2p for C and N, and 1$s$ for H. For the 2D perovskite structures, we adopted a $3 × 3 × 1$ Γ-centered k-point grid, generated by the Monkhorst–Pack scheme, for detailed properties obtained with PBE functional. Considering the interaction between the hydrogen atoms and high-electronegativity groups, the PBE with the DFT-D3 dispersion correction of Grimme with zero-damping was applied to optimize the geometric structures[49–51]. During the optimization of the geometries, all structures were allowed to relax to ensure that each atom was in mechanical equilibrium without any residual force larger than 10$^{-4}$ eV/Å.

## Reporting summary

Further information on research design is available in the Nature Portfolio Reporting Summary linked to this article.

## Data availability

The data generated in this study are provided in the Supplementary Information/Source Data file. Source data are provided with this paper.

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

## Acknowledgements

The work was supported by National Key Research and Development Program of China (No. 2023YFB3809700), Innovation and Technology Fund (GHP/100/20SZ, GHP/102/20GD, MRP/040/21X, ITS/147/22FP), Research Grants Council of Hong Kong Grant (N_CityU102/23, C4005-22Y, C1055-23G, 11306521), Green Tech Fund (GTF202020164), the Science Technology and Innovation Committee of Shenzhen Municipality (SGDX20210823104002015, JCYJ20220818101018038), National Natural Science Foundation of China (52322318). X.C.Z acknowledges the support by Hong Kong Global STEM Professorship Scheme.

## Author contributions

B.L., Q.L., J.G., S.L., and C.Z. contributed equally to this work. Z.Z. conceived the ideas, directed and supervised the research. B.L. conducted the investigation and characterization. J.G., S.L., and D.G. fabricated the devices. H.L. and Z.C. synthesized the MV materials. X.C.Z. and Q.L. conducted the DFT calculations. Z.L., X.W., D.Z., and Z.Y. were involved in device measurements and analysis. C.Z., X.L., and Y.W. were involved material characterization. B.L., Q.L., J.G., S.L., C.Z., and Z.Z. drafted and finalized the manuscript. All the authors revised the manuscript.

## Competing interests

The authors declare no competing interests.
