## [Peer Review File · Nature Communications]

Harnessing strong aromatic conjugation in low-dimensional perovskite heterojunctions for high-performance photovoltaic devicesEditorial Note: This manuscript has been previously reviewed at another journal that is not operating a transparent peer review scheme. This document only contains reviewer comments and rebuttal letters for versions considered at *Nature Communications*.

REVIEWERS' COMMENTS

Reviewer #1 (Remarks to the Author):

I have had a chance to read the response letter and the revised manuscript. I am satisfied with the revisions made and therefore I can recommend publication of this work in nature communications now.

Reviewer #2 (Remarks to the Author):

The authors addressed all the comments from each reviewer well. The MVPb2I6 used in this study has been previously documented in other research, with most references included in the revised version. Figure S17d of the energy level diagram should be included into the main figures of the revised manuscript to aid in comprehending electron extraction in the 1D/3D heterojunction between perovskite and C60. Therefore, I recommend that this manuscript be accepted in Nature Communications.

Reviewer #3 (Remarks to the Author):

The changes made in response to the reviewer's comments, the additional description of the experimental system and fabrication methods (lines 106-123), and the UPS-based discussion of the energy levels (lines 137 – 141), are acknowledged.

Please modify the text line 111 to include missing information “the characteristic peaks” [of XRD] and line 112 “surface morphology” [as determined by AFM measurements].

Other than this point, I think this paper is now ready for publication in Nature Commun.

REVIEWERS' COMMENTS

Reviewer #1 (Remarks to the Author):

I have had a chance to read the response letter and the revised manuscript. I am satisfied with the revisions made and therefore I can recommend publication of this work in nature communications now.

Reply: We thank the reviewer for the professional review and the recognition of our results.

Reviewer #2 (Remarks to the Author):

The authors addressed all the comments from each reviewer well. The MVPb_2I_6 used in this study has been previously documented in other research, with most references included in the revised version. Figure S17d of the energy level diagram should be included into the main figures of the revised manuscript to aid in comprehending electron extraction in the 1D/3D heterojunction between perovskite and C60. Therefore, I recommend that this manuscript be accepted in Nature Communications.

Reply: Thanks to the reviewer for recognizing our work and revisions, as well as providing us with valuable suggestion. We have included the Fig. S17d in Figure 1 in the main text. The corresponding Figure 1 have revised in manuscript.

Reviewer #3 (Remarks to the Author):

The changes made in response to the reviewer's comments, the additional description of the experimental system and fabrication methods (lines 106-123), and the UPS-based discussion of the energy levels (lines 137 – 141), are acknowledged.

Please modify the text line 111 to include missing information “the characteristic peaks” [of XRD] and line 112 “surface morphology” [as determined by AFM measurements]. Other than this point, I think this paper is now ready for publication in Nature Commun.

Reply: Thanks to the reviewer for acknowledging our work and revisions. For missing information of “the characteristic peaks” and “surface morphology”, we have added in the main text in the *structural characterization* section with highlighted in blue fonts. Moreover, we marked the characteristic peaks in XRD pattern in Figure S6 and S7.